# TILLING-by-Sequencing^+^ to Decipher Oil Biosynthesis Pathway in Soybeans: A New and Effective Platform for High-Throughput Gene Functional Analysis

**DOI:** 10.3390/ijms22084219

**Published:** 2021-04-19

**Authors:** Naoufal Lakhssassi, Zhou Zhou, Mallory A. Cullen, Oussama Badad, Abdelhalim El Baze, Oumaima Chetto, Mohamed G. Embaby, Dounya Knizia, Shiming Liu, Leandro G. Neves, Khalid Meksem

**Affiliations:** 1Department of Plant, Soil, and Agricultural Systems, Southern Illinois University, Carbondale, IL 62901, USA; naoufal.lakhssassi@siu.edu (N.L.); zhou1228@siu.edu (Z.Z.); mallory.cullen@siu.edu (M.A.C.); oussama.badad@gmail.com (O.B.); abdelhalim.elbaze@siu.edu (A.E.B.); oumaima.chetto@gmail.com (O.C.); dounya.knizia@siu.edu (D.K.); smliu@siu.edu (S.L.); 2Department of Animal Science, Food, and Nutrition, Southern Illinois University, Carbondale, IL 62901, USA; mohamed.embaby@siu.edu; 3RAPiD Genomics, Gainesville, FL 32601, USA; lneves@rapid-genomics.com

**Keywords:** TILLING by target capture sequencing, TILLING-by-sequencing^+^, oil biosynthesis pathway, fatty acid desaturases, *GmSACPD-C*, *GmFAD2-1A/1B*, *GmFAD3A/B/C*

## Abstract

Reverse genetic approaches have been widely applied to study gene function in crop species; however, these techniques, including gel-based TILLING, present low efficiency to characterize genes in soybeans due to genome complexity, gene duplication, and the presence of multiple gene family members that share high homology in their DNA sequence. Chemical mutagenesis emerges as a genetically modified-free strategy to produce large-scale soybean mutants for economically important traits improvement. The current study uses an optimized high-throughput TILLING by target capture sequencing technology, or TILLING-by-Sequencing^+^ (TbyS^+^), coupled with universal bioinformatic tools to identify population-wide mutations in soybeans. Four ethyl methanesulfonate mutagenized populations (4032 mutant families) have been screened for the presence of induced mutations in targeted genes. The mutation types and effects have been characterized for a total of 138 soybean genes involved in soybean seed composition, disease resistance, and many other quality traits. To test the efficiency of TbyS^+^ in complex genomes, we used soybeans as a model with a focus on three desaturase gene families, *GmSACPD*, *GmFAD2*, and *GmFAD3*, that are involved in the soybean fatty acid biosynthesis pathway. We successfully isolated mutants from all the six gene family members. Unsurprisingly, most of the characterized mutants showed significant changes either in their stearic, oleic, or linolenic acids. By using TbyS^+^, we discovered novel sources of soybean oil traits, including high saturated and monosaturated fatty acids in addition to low polyunsaturated fatty acid contents. This technology provides an unprecedented platform for highly effective screening of polyploid mutant populations and functional gene analysis. The obtained soybean mutants from this study can be used in subsequent soybean breeding programs for improved oil composition traits.

## 1. Introduction

Soybean [*Glycine max* (L.) Merr.] is the world’s largest oilseed crop and provides about 53% of vegetable oil in the U.S. [1]. Soybean oil has a wide range of utilization in human consumption, animal feeding, and industrial applications. Modification of the five major fatty acid content in soybean oil draws much attention, as well as the production of novel fatty acids for nutritional enhancement [2]. For many years, the soybean research community has been focused on the metabolic engineering of fatty acid biosynthesis pathways to genetically improve soybean oil composition traits using different approaches [3]. Fatty acid desaturases are a large group of important enzymes that control saturated/unsaturated fatty acid ratio in soybean seed. There are three major fatty acid desaturase gene families in soybeans, which are localized at either plastid or endoplasmic reticulum (ER).

In the plastid, delta-9-stearoyl-acyl carrier protein desaturase (SACPD) catalyzes the conversion of stearic acid to oleic acid [4]. Stearic acid (C18:0) typically represents 3–4% of total fatty acids in soybean commodities [5]. No negative effect on blood serum low-density lipoprotein (LDL) cholesterol has been associated with stearic acid content for human consumption. Stearic acid is also an essential oxidative stable component of soybean oil with a high melting temperature [6]. Rather than the hydrogenation process to produce *trans* fats, genetic manipulation of the fatty acid biosynthetic pathway is the most efficient approach to elevate stearic acid content in soybean seed oil. Mutations at the seed-specific *GmSACPD-C* gene resulted in a range of 6–20% stearic acid content in soybean seeds [7,8,9,10,11]. A dual role of *GmSACPD-C* in both soybean unsaturated fatty acid biosynthesis and nitrogen-fixing nodules has been reported [11,12]. Deleterious mutations at conserved residues of GmSACPD-C have been confirmed to cause the atypical nodule structure, while healthy nodules were observed for non-deleterious mutations (soft mutations) at non-conserved residues [11].

The omega-6 fatty acid desaturase 2 (FAD2) converts oleic acid into linoleic acid in the fatty acid biosynthesis pathway (Appendix A) [13]. Oleic acid (18:1, ω-9) is a monounsaturated fatty acid that represents ~18–20% content in conventional soybean oil [14]. The elevation of oleic acid content through chemical hydrogenation has been employed to improve oxidative stability and shelf life of soybean oil. However, such a process would cause human heart-related problems [15]. Two *GmFAD2-1* genes have been characterized in the soybean genome, including *GmFAD2-1A* (Glyma.10G278000) and *GmFAD2-1B* (Glyma.20G111000) [16,17]. The two microsomal *GmFAD2-1* desaturase genes possess the highest seed-specific expression level and control the oleic acid content in soybean seeds [18]. The association of *GmFAD2-1A/1B* and oleic acid content has been revealed in both segregating and mutagenized soybean populations [19,20]. A combination of *GmFAD2-1A* and *GmFAD2-1B* mutant alleles have been reported to achieve more than >80% oleic acid content in soybean seeds [21,22,23]. Several technologies have been used to considerably improve oleic acid content, including gene-editing technology, Transcription activator-like effector nucleases (TALENs), and clustered regularly interspaced short palindromic repeats (CRISPR-Cas9) [24,25].

The microsomal omega-3 fatty acid desaturase (FAD3) converts linoleic acid to linolenic acid in the fatty acid biosynthetic pathway. Polyunsaturated fatty acids, including linoleic acid (18:2, ω-6) and α-linolenic acid (ALA, 18:3, ω-3), typically constitute 60% of conventional soybean oil [14]. High polyunsaturated fatty acid content in soybean oil has low oxidative stability and must be hydrogenated for many applications [26]. Within the soybean genome, four *GmFAD3* genes have been previously identified as *GmFAD3A* (Glyma.14g194300), *GmFAD3B* (Glyma.02g227200), *GmFAD3C* (Glyma.18g062000), and *GmFAD3D* (Glyma.11g174100) [27,28]. A series of novel allelic variations in *GmFAD3A* have been reported as a source that reduces linolenic acid content [29,30]. Reinprecht et al. [31] isolated a low linolenic acid mutant line (RG10) resulting in a truncated GmFAD3A protein and a splicing mutation at the GmFAD3B. Other novel mutations at the *GmFAD3C* gene have also been identified and resulted in low linolenic acid content in soybean seeds [32,33]. Soybean lines containing 1% linolenic acid in the seed oil have been developed when combining mutations in the three *GmFAD3* genes [34,35]. By stacking *GmFAD2s* and *GmFAD3s* mutations, non-transgenic soybean lines with low linolenic acid content (<3%) and extremely high oleic acid content (>80%) have been achieved [36,37].

As an efficient reverse genetic method in functional genomics; TILLING (Targeting Induced Local Lesions IN
Genomes), was used to identify induced mutations from a mutagenized population. It typically entails chemical mutagenesis and a high-throughput mutation screening method. Ethyl methanesulfonate (EMS) is regarded as the most commonly used chemical mutagen to randomly create point mutations in plant genomes [38,39]. In 2008, conventional TILLING was employed in soybean to screen for induced mutations in seven genes among four mutagenized soybean populations [40]. However, the duplicated genome from which several homologous genes controlling one trait could impede mutation discovery in soybeans [17]. Although a large size of mutagenized soybean populations has been developed, the number of identified mutants in targeted genes is scarce using gel-based TILLING and forward screening. Given the dramatic decline of sequencing cost, next-generation sequencing (NGS) technologies have been integrated into the TILLING pipeline for mutation discovery since 2009 [41]. Using TILLING by
Sequencing (TbyS), several recent studies have been reported to improve the efficiency of mutation detection from polyploid species, including durum wheat (allotetraploid), rice (diploid), soybean (palaeopolyploid), crambe (hexaploid), and peanut (allotetraploid) [42,43,44,45,46]. A variety of bioinformatic tools have also been developed and evaluated for SNP calling [47,48].

Here, we report the use of TILLING-by-Sequencing^+^ (TbyS^+^) technology and its application in identifying induced mutations over 138 genes, including members of three key soybean fatty acid desaturase gene families. A comprehensive analysis of the three fatty acid desaturase gene families was conducted including gene evolution, gene duplication, gene structure, and expression profile. An extra-large number of EMS-induced mutagenized soybean populations have been developed (4032 M2 families) as a diverse genetic resource for economically important traits. The soybean mutants carrying mutations at the six fatty acid biosynthetic genes were phenotyped for their fatty acid profiles. The identified new alleles could be used in soybean breeding programs for improved oil composition traits.

## 2. Results

### 2.1. Soybean Mutant Library Construction

A total of 4032 M2 families have been selected to establish a mutant library for TbyS^+^. Young leaves were collected from independent M2 plant, and then genomic DNA were extracted and quantified. A total of 42 plates comprising 4032 DNA samples were pooled into three plates (P1, P2, and P3) using a bidimensional-arraying strategy. From 168 vertical pools in P1 and P2, each pool contained 24 DNA samples, whereas 48 DNA samples were pooled into each of the 84 horizontal pools in P3, for high-throughput mutation screening (Appendix A). The M3 seeds harvested from individual M2 plants were stored at −20 °C for further seed phenotyping analysis.

### 2.2. Mutant Retrieval from TILLING by Sequencing^+^ (TbyS^+^)

A total of 138 soybean genes were selected to design probes for amplicon sequencing (Appendix A). At least two genes were selected from each of the 20 soybean chromosomes, with a minimum of two genes on chromosome 12 and a maximum of 11 genes on chromosome 10. Disease resistant and seed composition-related genes accounted 41% and 36% of the selected 138 genes, respectively. While the rest of genes were selected based on other important soybean traits, such as abiotic stress tolerance, plant hormone signaling, plant development, and epigenetics. To ensure the specificity of the amplifications, a series of probes were designed to cover all exons of each gene for NGS. For fatty acid desaturase genes, 41 probes were constructed to amplify the whole exon regions of *GmSACPD-C* with 97.4% coverage. For *GmFAD2-1A* and *GmFAD2-1B*, 44 and 77 probes were designed with 99.6% and 97.4% coverage, respectively. *GmFAD3A*, *GmFAD3B*, and *GmFAD3C* were covered by 48, 52, and 53 probes with 91.0%, 97.4% and 94.0% coverage (Appendix A). The pooled amplicon library was then sequenced to obtain approximately 450 million reads from three lanes of Illumina HiSeq X. Prior to SNP calling, the sequencing data were processed through an automatic workflow consisting of BWA and SAMtools using the parallel function. 

The sorted BAM files were then used to run Comprehensive Read Analysis for the Identification of SNVs (and short InDels) from Pooled sequencing data (CRISP) for SNP discovery. The original Variant Call Format (VCF) files were filtered through SnpSift and stats of SNPs for each gene were visualized using Integrative Genomics Viewer (IGV) (Appendix A). For each mutation, CRISP provided a list of descriptions, including type of mutation, position, allele frequency (AF), quality score (QUAL), and alt allele counts (AC). Based on the QUAL and AC of all previously identified mutations, the filtering condition consisting of QUAL equal to 700 or higher and AC equal to 4 or higher has been set as a conservative threshold to obtain true mutations. The same mutation in one gene was identified in one well from vertical pools and another well from horizontal pools. Two candidate mutant lines were then determined to contain this mutation through demultiplexing the vertical and horizontal pools in these two wells. In the last step, genomic DNA from these two mutant lines was retrieved for Sanger sequencing to confirm the mutation in one of these two mutant lines (Figure 1).

### 2.3. Characterization of Induced Mutations over 138 Soybean Genes Identified through TbyS^+^

The number of mutations identified by TbyS^+^ were summarized for each of the 138 soybean genes and organized under each of the 20 soybean chromosomes (Figure 2). The typical EMS-type mutations, G/C to A/T, were found to be dominant in all identified induced mutations for targeted genes on each chromosome (Figure 2A). Notably, 82.5% of induced mutations belong to the EMS-type mutations, including 42.4% of G to A and 40.1% of C to T, while the other types of base changes only account for 17.5% within the 138 soybean genes (Figure 2B). The overall mutation density for the 138 genes is estimated at 1/227 kb (Appendix A).

The effect of induced mutations in the coding region of each gene was also analyzed using Variant Effect Predictor (VEP) program at Ensembl Plants (Figure 3). The majority of mutations in the coding regions resulted in missense and silent mutations for targeted genes on each chromosome while few other genes possessed nonsense mutations with maximum number of 6 (Figure 3A). For all 138 genes, the percentages of missense, silent, and nonsense mutations were 65.5%, 30.4%, and 4.1%, respectively (Figure 3B). Based on the induced mutations in the six fatty acid desaturase genes identified through TbyS^+^, we discovered 274 SNP mutations, including 111 of G to A, 107 of C to T, and 56 corresponding to other SNP mutations.

About 79.6% of SNP mutations are the EMS-type mutations (G/C to A/T) while the other types of mutations account for 20.4% of the total base changes. The mutation density for the five fatty acid desaturase genes is estimated as 1/212 kb (Appendix A). *GmFAD2-1B* is the only gene whose number of another type of mutation is larger than the number of either G to A or C to T (Table 1). Within the coding regions of six fatty acid desaturase genes, we observed a total of 147 amino acid changes, from which the missense, silent, and nonsense mutations were 92, 50, and 5, respectively (Appendix A). Several nonsense mutations were found in *GmFAD2-1B* (1), *GmFAD3B* (3), and *GmFAD3C* (1) (Table 1).

### 2.4. TbyS^+^ for Rescreening the Putative Mutations

To test the performance of TbyS^+^, we selected a total of 12 previously identified mutants in *GmSACPD-C* [11], and *GmFAD2-1A/1B* [17] genes as positive controls to rescreen in reads produced by target sequencing. Based on the corresponding mutant line, the two wells from vertical and horizontal pools were recovered, and the type of base change and SNP position were utilized to confirm the mutation in each gene. Evidently, 7 out of 10 mutants were able to recover through TbyS^+^, including four *GmSACPD-C* mutants (C305T in F813, G229A in F714, C235T in F620, and G340A in F869), and three *GmFAD2-1A* mutants (C301T in F1235, C851T in F1284, and C745T in F1274) [11] (Appendix A). The TbyS^+^ technology provides a high success rate for true mutation detection, 4/5 (80%) for *GmSACPD-C* and 3/4 (75%) for *GmFAD2-1A*.

### 2.5. Identification of Novel Alleles of Fatty Acid Biosynthetic Genes Using TbyS^+^

Using TbyS^+^, a series of novel mutants were identified from six fatty acid desaturase genes, including *GmSACPD-C*, *GmFAD2-1A/1B*, and *GmFAD3A/B/C* (Figure 4). The forward genetic screening revealed the presence of altered fatty acid phenotypes from a variety of selected mutants.

S48N, R108W, G185E, A238V, G244R and G261D mutations were detected at *GmSACPD-C*, from which two mutations (G143A and C322T) were at the first exon and the other four (G554A, C713T, G730A and G782A) were located at the second exon (Appendix A). A threefold increase in stearic acid content was observed in the F2146 mutant (Table 2).

A total of 16 novel missense/nonsense mutations were identified from *GmFAD2-1A* (P30S, G39D, L100F, K168Q, E225K and P354S) and *GmFAD2-1B* (H113P, R147H, V169I, S232F, P262S, A282V, M328I, H332Y, E377K, and W382*) (Appendix A). Four *FAD2-1A* mutants showed increased oleic acid content (25.0–34.5%), in which the F258 mutant contained the highest level of oleic acid (34.5%). The oleic acid contents of five *FAD2-1B* mutants ranged between 23.0% and 27.8% (Table 2).

Six novel mutations were confirmed in each of the three omega-3 fatty acid desaturase genes, from which 16 mutants carried missense mutations (P28S, P171S, G277D, L279F, D352N, and D360N at GmFAD3A; P32S, P154L, G282D, H306Y, and A323T at GmFAD3B; A38T, S112N, G128E, R285H, and Q333P at GmFAD3C) and another two carried nonsense mutations (W247* at GmFAD3B and W267* at GmFAD3C) (Appendix A). The linolenic acid content of 11 *GmFAD3* mutants ranged from 4.5% to 5.8% compared to Forrest wild type (7.2%). Among them, one *GmFAD3A* mutant (F1012) presented the lowest linolenic acid content (4.5%), and two *GmFAD3B* mutants (F475 and F728) contained 4.6% linolenic acid content (Table 2).

### 2.6. Detection of New Conserved Residue within the Fatty Acid Desaturase Genes

Multiple fatty acid desaturase protein sequence alignments were conducted to investigate whether the positions of selected novel mutations are contained in conserved regions of the fatty acid desaturase genes. The GmSACPD-C, GmFAD2-1A/1B, and GmFAD3A/B/C protein sequences were aligned with their orthologous proteins from other eight plant species, including a dicot model (*A. thaliana*), three legume species (*P. vulgaris*, *M. truncatula* and *L. japonicas*), two dicot species (*L. usitatissimum* and *O. europaea*), and two monocot species (*E. guineensis* and *O. sativa*). Three mutations at the SACPD-C were located at highly conserved residue positions (R108, G185, and G244) and one at A less conserved residue position, G261 (Figure 5).

Unlike the F1320 (SACPD-C_G261D_), F2146 (SACPD-C_R108W_), F186 (SACPD-C_G185E_), and F1202 (SACPD-C_G244R_) belonged to conserved residues group. Out of the four mutations identified at the *GmFAD2-1A*, three mutations (F1356, F765, and F101) were found at conserved residues (FAD2-1A_P30S_, FAD2-1A_G39D_, and FAD2-1A_E225K_), while only one mutation (F258) was found at non-conserved residues (FAD2-1A_K168Q_) (Figure 6). Three non-conserved residues (FAD2-1B_V169I_ in F782, FAD2-1B_A282V_ in F36, and FAD2-1B_E377K_ in F903) and two conserved residues (FAD2-1B_H113P_ in F966 and FAD2-1B_P262S_ in F720) at the *GmFAD2-1B* were identified (Figure 6).

Moreover, most of the eleven mutations at the *GmFAD3-A/B/C* were located in highly conserved residue positions except two (WX247 and EX333) (Figure 7). All four *GmFAD3A* mutations were at conserved residue positions, including F1178 (FAD3A_P171S_), F180 (FAD3A_G277D_), F1012 (FAD3A_L279F_), and F1428 (FAD3A_D360N_). Besides the F560 (FAD3B_W247*_) nonsense mutant, the other three *GmFAD3B* mutations, F475 (FAD3B_P154L_), F728 (FAD3B_G282D_), and F461 (FAD3B_H306Y_), were located at conserved residues. Two conserved residues (FAD3C_A38T_ in F953 and FAD3C_G128E_ in F846) and one non-conserved residue (FAD3C_Q333P_ in F1739) were identified from three *GmFAD3C* mutations (Figure 7).

### 2.7. Phylogenetic Analysis of Fatty Acid Desaturases in Soybean

The fatty acid desaturases are constituted of at least three gene families in soybean, including the stearoyl-acyl carrier protein desaturase (*GmSACPD*), omega-6 fatty acid desaturase (*GmFAD2*), and omega-3 fatty acid desaturase (*GmFAD3*, *GmFAD7*, and *GmFAD8*). Previous studies described three actively transcribed members of the *GmSACPD* gene family and seven members of *GmFAD2* gene family in soybean [7,17]. Soybean omega-3 fatty acid desaturase gene family contains four members in the ER and two in the plastid. A total of 19 soybean fatty acid desaturases have been adopted for phylogenetic analysis. A ML tree was built with 90 protein sequences to elucidate the phylogenetic relationships among fatty acid desaturases from nine plant species, including four legume species, three dicot species, and two monocot species (Figure 8). As expected, the clade containing the SACPD grouped separately from the FAD2 and FAD3. Considering the high saturated fatty acid content in palm oil, it was consistent to find the EgSACPD (EGU1685G2068) in the same sub-clade, including the GmSACPD-C/D, while the other three EgSACPD members were grouped with the GmSACPD-A/B. The GmFAD2-1A/1B and GmFAD2-2A/B/C/E were separately grouped into two sub-clades, except for GmFAD2-2D that formed a new sub-clade. Interestingly, given the high oleic acid content in seed, all five OeFAD2 from olive (*O. europaea*) clustered apart from GmFAD2 (Figure 8). Four microsomal omega-3 desaturases, GmFAD3A/B/C/D, clustered together in one sub-clade while another sub-clade contained the plastidial GmFAD7 and GmFAD8. The two LuFAD3 (Lus10038321 and Lus10036184) from flax (*L. usitatissimum*), which is well known to accumulate ALA, clustered in the same sub-clade with the soybean microsomal omega-3 desaturases (Figure 8).

The coding sequence (CDS) lengths of the four *GmSACPD* members varied from 1014 bp to 1209 bp (*GmSACPD-A*) with an average of 1134 bp. The average CDS length of the five *GmFAD2* members was 1152 bp while it was limited to only 651 bp and 882 bp for the other two members, *GmFAD2-2A* and *GmFAD2-2E*, respectively. The average CDS length of the four microsomal *GmFAD3* was 1140 bp, which is 221 bp shorter than that of the plastidial *GmFAD7* and *GmFAD8* (1361 bp) (Appendix A). Regarding the physical properties of the fatty acid desaturase proteins, the length and predicted molecular weight of GmSACPD proteins ranged from 337 to 402 amino acids and 38.6 kDa to 46.1 kDa, respectively. The GmFAD2 proteins consisted of 216 to 387 amino acids with 25.5 kDa to 44.7 kDa in molecular weight. The average length and predicted molecular weight of the GmFAD3 and GmFAD7/GmFAD8 proteins was 379 amino acids (44.0 kDa) and 453 amino acids (51.3 kDa), respectively. GmSACPD possessed an acidic isoelectric point (pI); however, the other four fatty acid desaturases (GmFAD2, GmFAD3, GmFAD7, and GmFAD8) showed a basic pI value (Appendix A).

### 2.8. Gene Structural and Tissue-Specific Expression Profiling of Fatty Acid Desaturases in Soybean

Due to the two whole-genome duplication events that occurred in the soybean genome, at least 19 members constitute the fatty acid desaturases in soybean, which is a two-fold increase in Arabidopsis, common bean, and rice. *GmSACPD-A/B* subfamily possessed three exons while the *GmSACPD-C/D* subfamily carry two exons. The overall gene length of *GmSACPD-A/B* was also larger than that of *GmSACPD-C/D* due to their extended intron length (Figure 9). The *GmFAD2-1* gene family carried two exons with similar gene lengths (1164 bp) (Figure 9). The gene structure was highly conserved with eight exons across all the *GmFAD3*, *GmFAD7*, and *GmFAD8* subfamilies. Although *GmFAD3A* and *GmFAD3B* possessed larger gene sizes, the CDS lengths of *GmFAD3* were shorter than that of *GmFAD7/GmFAD8* (Figure 9; Appendix A).

Next, expression profiling of the 14 soybean fatty acid desaturases was carried out among seven soybean tissues. *GmFAD2-1* subfamily presented remarkably high expression levels in the late stages of seed development, leaf, flower, and pod. For the omega-3 desaturase gene family, except *GmFAD3D*, the rest of the *GmFAD3* members showed fairly high expression in seeds, while the transcripts of two *GmFAD8* genes were abundant in leaf. *GmFAD3B* and *GmFAD3C* were also highly expressed in leaf and pod. In contrast, two *GmFAD7* genes were expressed at drastically low levels in all tissues except pod shells. A low expression level of all eight omega-3 desaturase genes was detected in root and nodule (Appendix A).

### 2.9. Chromosomal Distribution and Syntenic Analyses of Fatty Acid Desaturases in Soybean

Nineteen soybean fatty acid desaturase genes were unevenly distributed on 13 chromosomes, from which two genes were present on chromosomes Chr02, Chr03, Chr07, Chr14, Chr18, and Chr19 and one gene was on chromosomes Chr01, Chr09, Chr10, Chr11, Chr13, Chr15, and Chr20 (Figure 10). The four *GmSACPD* gene family members were located at Chr07 (*GmSACPD-A*), Chr02 (*GmSACPD-B*), Chr14 (*GmSACPD-C*), and Chr13 (*GmSACPD-D*). Six chromosomes contained the seven *GmFAD2* gene family members, including *GmFAD2-1A* on Chr10, *GmFAD2-1B* on Chr20, *GmFAD2-2A* and *GmFAD2-2B* on Chr19, *GmFAD2-2C* on Chr03, *GmFAD2-2D* on Chr09, and *GmFAD2-2E* on Chr15. Similarly, eight soybean omega-3 desaturase genes were distributed on seven chromosomes, in which *GmFAD3A* was located on Chr14, *GmFAD3B* on Chr02, *GmFAD3C* and *GmFAD7-1* on Chr18, *GmFAD3D* on Chr11, *GmFAD7-2* on Chr07, *GmFAD8-1* on Chr01, and *GmFAD8-2* on Chr03 (Figure 10).

In the soybean genome, nine duplicated blocks containing 18 fatty acid desaturase genes were identified through the plant genome duplication database (PGDD), including eight segmental and one-tandem duplication blocks (*GmFAD2-2A* and *GmFAD2-2B*) (Figure 10; Appendix A). The ratios of nonsynonymous to synonymous substitutions (Ka/Ks) were calculated for each gene pair to determine the types of natural selection acting on their corresponding coding sequences. The Ka/Ks of all nine gene-pairs were less than 1, which suggests that the evolution of these fatty acid desaturase genes is under purifying selection [49,50]. The duplication of the nine gene-pairs was estimated to have occurred between 7.38 and 106.56 Mya based on 6.161029 synonymous mutations per synonymous site per year for soybean.

## 3. Discussion

Soybean is a leading oilseed that is grown worldwide. Improving seed oil composition traits is at the core of soybean breeding. However, traditional breeding is labor-consuming, and therefore, it hindered the rapid development of new soybean germplasm into the market. Currently, the major source of improved oil/fatty acid composition traits was mainly produced by genetically modified crops. Such products must comply with the restrictive regulations and cause solicitude from consumers. To avoid health risks and meet nutritional needs, numerous efforts have been made to produce soybeans with elevated oleic acid and low polyunsaturated fatty acid contents [36,37]. In this study, a considerable number of soybean mutants with altered fatty acid contents are leveraged across six major fatty acid desaturase genes using large-scale mutation breeding. Combined with novel *GmSACPD-C*, *GmFAD2-1A/1B*, and *GmFAD3A/B/C* alleles, the goal of developing soybean with high stearic acid, high oleic acid and low polyunsaturated fatty acid contents can be achieved under the same genetic background in a timely manner.

In the past decade, there were tremendous efforts to develop functional analysis tools in soybean. RNAi and virus-induced gene silencing have been extensively used to study soybean genes. However, gene silencing technology lacked precision due to the similarity between the soybean gene family members. Its application also largely depends on tissue and time specificity expression. Therefore, there is a strong need to develop a new gene functional analysis tool. Coupled with chemical mutagenesis, the high-throughput screening techniques have increasingly progressed to meet the needs of functional genomic studies in major crops. Here, we presented a target capture method facilitated by NGS to efficiently identify population-wide induced mutations in genes controlling economically important traits. In this study, the size of screening mutant populations (4032 M2 families) and the number of targeted genes (>138) were far more than previous studies [41,45,51]. Moreover, considering the cost-effectiveness, the newly developed TbyS^+^ technology maximized the multiplexing in the mutant library without sacrificing the accuracy of detecting mutations [52]. Instead of eight individual samples in one pool, 24 and 48 individual samples were pooled together to dramatically decrease the number of pools for sequencing (Appendix A). The series of probes were designed with high coverage for the whole region of targeted genes (Appendix A).

Furthermore, right after DNA library preparation, the use of capture-seq-enrichment facilitated by target recovery technology before moving to the NGS considerably increases the designed probes specificity and the number of true positive mutants obtained. This step eliminated the issues related to gene families, the gene with high copy number and similarities within the soybean genome, and provided a robust solution to isolate mutants in complex and duplicated genomes. Using parallel commands, all raw sequencing data were automatically processed before SNP calling. Due to the complexity of multiplexing, *freebayes*, a haplotype-based variant detector, failed to call mutations in our initial SNP calling test [53]. 

After switching to CRISP, we were able to detect SNPs and small InDels from sequencing pooled samples using the Illumina platform. Based on the quality score (QUAL) and Alt Allele Counts (AC), the filtering retained up to 92.3% of novel mutations but removed the majority of spurious ones when applying such thresholds (QUAL ≥ 700 and AC ≥ 4). Although the individual mutant line was not revealed directly after SNP calling, the number of lines with the targeted mutation would be confined to only two candidates, and the mutant can be eventually determined by Sanger sequencing, which is still the most reliable way of mutation validation [41,43]. Our method provided an inventory containing the largest dynamic allelic series of mutations within target genes and represented the most powerful tool for functional genomic studies in soybean to date.

Ethyl methanesulfonate (EMS) is the most commonly used chemical mutagen to randomly create point mutations in plant genomes [38,39]. Most of the mutations that were present in the oil biosynthesis genes due to the EMS mutagenesis are SNPs, with G > A and C > T being the most type of base changes that are present at highest percentages: 91%, 88%, 62%, 77%, 88%, 75% for *GmSACPD-C*, *GmFAD2-1A*, *GmFAD2-1B*, *GmFAD3A*, *GmFAD3B*, and *GmFAD3C*, respectively. Unlike other mutagen agents (like sodium azide or fast neutron), EMS mutagenesis does not generate large deletions. At the M2 generation, most of the mutations are heterozygous. Therefore, the plants are advanced to the M3 generation, where, 25% revert to the wild type, 50% are heterozygous, and 25% are homozygous for the desired mutation. The M3 mutants are sanger sequenced to validate the homozygous mutations for further phenotyping.

The remarkable accomplishment of TbyS^+^ is justified by confirming our published mutations in genes controlling the fatty acid composition and disease resistance traits [11,17,54,55,56,57]. Only three out of 12 fatty acid desaturase mutants were not detected using this method. The possible reason could be the slight difference between the reference genome Williams 82 and Forrest for target genes and may also be due to the experimental error during DNA extraction. The estimated mutation density in this study (~1/227 kb) was found to be within the range as described from previous reports in soybean (between ~1/140 kb and ~1/550 kb) [40]. The percentage of the EMS-type mutations (G/C to A/T) is close to ones found in rice and barley (>70%) but lower than ones reported in maize and wheat (>99%) [58]. Using TbyS^+^, a variety of novel missense/nonsense mutations in fatty acid desaturase genes within the fatty acid biosynthesis pathway have been identified. Besides the five mutations that were previously identified in the first exon of *GmSACPD-C* [11], four novel missense mutations were detected in the second exon (Appendix A). The additional nine novel missense mutations and one nonsense mutation were also isolated at *GmFAD2-1B* as well as six novel missense mutations in *GmFAD2-1A*. From 18 newly identified *GmFAD3* mutations, one nonsense mutation each was detected at *GmFAD3B* and *GmFAD3C* (Appendix A).

The novel identified mutations within the six fatty acid desaturase genes at the fatty acid biosynthesis pathway present novel sources to improve soybean oil composition traits. Within four *GmSACPD-C* mutations, three mutations located at conserved residue positions (R108W, G185E, and G244R) have shown a greater impact on stearic acid content (Figure 5; Table 2). R108W is speculated to deeply affect SACPD-C enzyme activity due to its location at the di-iron center (Appendix A). Interestingly, the *GmFAD2-1A* mutation in non-conserved residue (K168Q) presented a higher oleic acid content than other three *GmFAD2-1A* mutations in conserved residues.

However, the difference in oleic acid content is trivial between conserved and non-conserved residues for *GmFAD2-1B* mutations (Figure 6; Table 2). Although the truncated W247* nonsense mutations was detected at GmFAD3B, the dramatic decrease in linolenic acid content was not observed because *GmFAD3A* was reported to have a greater impact on linolenic acid content and the highest expression levels in seeds among three *FAD3* genes in soybean [59]. The *GmFAD3A/B/C* mutations at conserved residues are all found to be associated with low levels of linolenic acid (Figure 7; Table 2). Although forward genetics (phenotyping) is an important tool to analyze soybean seed composition traits, the whole process to associate altered phenotypes with causal mutations is lengthy due to the presence of various environmental factors that could impact the observed phenotypes. As a newly emerged reverse genetic method, TbyS^+^ is a revolutionary approach to obtain an allelic series of mutations in candidate genes from large-scale mutant populations. This method can also be easily applied in other crop species using various mutant populations. The availability of mutations could be a valuable resource for molecular breeding.

Fatty acid desaturases comprising three desaturase families are the major component in the soybean fatty acid biosynthesis pathway. Previous studies largely focused on the identification and functional characterization of single fatty acid desaturase gene family. In this study, we performed a comprehensive phylogenetic analysis of the fatty acid desaturase gene families across nine plant species, including other three legume species and plant species with significant oil composition traits, such as palm (elevated saturated fatty acids level), olive (high oleic acid content), and flax (high polyunsaturated fatty acids level) (Figure 8). A large number of soybean fatty acid desaturase genes implied genome expansion of the soybean compared to their counterparts in other plant species. The phylogenetic analyses provided convincing evidence to support the observed differences in their corresponding subcellular localization between SACPD and FAD2/FAD3 [60]. It also showed that two subfamilies of *SACPD* have evolved independently to acquire distinct functions (Figure 8). One *SACPD* gene subfamily member played an essential role in the conversion of stearic acid to oleic acid while another one may be involved in plant defense mechanism against pathogens [61]. The gene structure and expression profiles undoubtedly reflected the striking difference between two *GmSACPD* subfamilies in soybean (Figure 9; Appendix A). 

Within the *FAD2* gene family, FAD2 from four legume species showed a close evolutionary relationship besides other five plant species. FAD2 from monocot species are more likely to be the ancestors for the rest of FAD2 (Figure 8). The unique gene structure of *GmFAD2-1* and their significant high expression in soybean seed demonstrated that only *GmFAD2-1* catalyzed the conversion of oleic acid into linoleic acid. The *GmFAD2-2* subfamily that clustered in a different sub-clade may gain novel function in the plant cell [17]. Among nine plant species, the evolution of microsomal and plastidial omega-3 desaturases was independent but originated from *Arabidopsis AtFAD3* (AT2G29980). Although soybean microsomal and plastidial omega-3 desaturases were grouped in different sub-clades, they presented a highly conserved gene structure and catalytic residues in their structural motifs (Figure 7 and Figure 9). Similar expression patterns within microsomal or plastidial omega-3 desaturase genes pointed to functional redundancy during soybean evolution, which could lead to either neofunctionalization or subfunctionalization within the omega-3 desaturase gene family (Appendix A).

Nineteen soybean fatty acid desaturase genes widely spread on thirteen chromosomes with a maximum of two genes on a chromosome. The positions of fatty acid desaturase genes were mostly towards the chromosome ends with high genetic recombination rates (Figure 10). Plant species acquired novel traits and adapted to various environments through gene duplication [62,63]. There are three main gene duplication patterns, including segmental duplication, tandem duplication, and transposition [64]. The syntenic analysis suggested that segmental duplications may play an essential role in the expansion of fatty acid desaturase genes in soybean (Appendix A). Two whole-genome duplication events have occurred in soybean genome, including one shared by legume species 59 Mya and another glycine-specific one around 13 Mya [65,66,67]. The duplication time of six fatty acid desaturase gene pairs were estimated to match with the late duplication event, including *GmSACPD-A/GmSACPD-B*, *GmFAD2-1A/GmFAD2-1B*, *GmFAD3A/FAD3B*, *GmFAD3C/FAD3D*, *GmFAD7-1/FAD7-2*, and *GmFAD8-1/FAD8-2*. The tandem duplication of *GmFAD2-2A*/*GmFAD2-2B* was estimated about 25.41 Mya ago while other two segmental duplications, *GmSACPD-C*/*GmSACPD-D* and *GmFAD2-2C*/*GmFAD2-2D* may have occurred 36.07 and 106.56 Mya ago (Appendix A).

## 4. Materials and Methods

### 4.1. Plant Materials, Growth, and EMS Mutagenesis

EMS mutagenesis was performed as previously described [68]. The soybean cv. Forrest and PI88788 seeds were used to generate M2 population using EMS in the greenhouse at the Horticulture Research Center, Southern Illinois University Carbondale. A total of 4032 M2 lines were advanced to the M3 generation by single-seed descent in the field between 2012–2015 [11,17,50]. M3 seeds from M2 mutant plants were harvested, threshed, and stored at −20 °C.

### 4.2. DNA Extraction and Quantification

Young leaves from 4032 M2 plants were collected and stored at −80 °C for DNA preparation. Leaf tissue (50–100 mg) was disrupted with tungsten carbide beads in a 96-well plate using TissueLyser System (QIAGEN), and DNA was extracted using DNeasy 96 Plant Kit (QIAGEN, Valencia, CA, USA). The quantity of DNA was estimated using Synergy 2 Multi-Mode Microplate Reader (BioTek Instruments Inc., Winooski, VT, USA). The concentration of each DNA sample was normalized to 100 ng/uL. All DNA samples in the 96-well plates were stored at −20 °C for long-term purposes.

### 4.3. Library Preparation, Probe Design and TbyS^+^

Genomic DNA samples from 96-well plates were pooled using bidimensional-arraying strategy to increase the screening throughput. In vertical pools, DNA arrayed from each of two plates were placed vertically in one column of P1 or P2 plates. Each well in the P1 and P2 plates contains 24 samples from the original DNA plates. In horizontal pools, DNA arrayed from the same row of each six plates were pooled in one row of P3 plate, and 48 samples were contained in each well of the P3 plate. The soybean-targeted capture design was developed on the capture-seq platform at Rapid Genomics (Gainesville, FL, USA) (Figure 1). The custom probes were designed and synthesized to cover the exons of each targeted gene. DNA libraries and capture enrichment were automatically prepared in-house. One hundred and fifty bp paired-end reads were generated on three lanes of Illumina HiSeqX with 10× sequencing depth per haploid genome.

### 4.4. Variant Calling for Mutation Detection

The FASTQ raw reads were subjected to quality control using FastQC v0.11.9, while trimming and filtering of low-quality reads were performed using Trimmomatic V0.39 [69]. The clean FASTQ reads were mapped to Williams 82 (Wm82.a2.v1) reference genome using BWA-0.7.17 [70]. SAM tools v1.10 [52] were used to filter and sort the BAM files to serve as an input for variant calling using Freebayes [71] and CRISP v1.18.0 [72]. The VCF files were filtered by VCF tools v0.1.16 [51] and visualized in IGV v 2.9.2 [73].To retain the true induced mutations, VCF files generated by CRISP were filtered by SnpSift and visualized through IGV for demultiplexing [73,74]. The effect of mutations in each gene was predicted using Variant Effect Predictor (VEP) program at Ensembl Plants release 45 [75]. The reference genome WI82.a2.v1 was used for SNP calling and mutation effect prediction. The isolated TILLING mutants have been target sequenced using Sanger sequencing to confirm the mutations.

### 4.5. Mutation Validation

Sanger sequencing was used to confirm all identified mutations at the three fatty acid desaturase gene family members (Appendix A). PCR(Polymerase Chain Reaction) primers were designed to amplify the fragments covering the exons of the six fatty acid desaturase genes, including *GmSACPD-C*, *GmFAD2-1A*, *GmFAD2-1B*, *GmFAD3A*, *GmFAD3B*, and *GmFAD3C* using Primer3 [76]. The PCR program was set up with 30 cycles of amplification at 94 °C for 30 s, 52 °C for 30 s, and 72 °C for 1 min. The PCR products were then purified using QIAquick Gel Extraction Kit (QIAGEN). The purified samples were sent for Sanger sequencing at GENEWIZ (https://www.genewiz.com/). Putative mutations were identified by alignment of sampled sequences to the reference genome using Unipro UGENE [77].

### 4.6. Fatty Acid Analysis of Mutant Seeds

Five major fatty acid contents were measured from individual seeds of mutant lines according to the two-step methylation procedure [78]. At least three seeds per line were individually crushed in 16 mm × 200 mm tubes with Teflon-lined screw caps. Two mL of sodium methoxide was added into each tube followed by 50 °C incubation for 10 min. After 5 min of cooling, the samples were mixed with 3 mL of 5% (*v*/*v*) methanolic HCl, incubated at 80 °C for 10 min, and cooled for 7 min. 7.5 mL of 6% (*w*/*v*) potassium carbonate and 2 mL of hexane were added to each tube and centrifuged at 1200× *g* for 5 min. The upper layers were transferred to vials, from which the individual fatty acid contents were determined as a percentage of the total fatty acid content in soybean seed by gas chromatography. A Shimadzu GC-2010 (Shimadzu Co., Kyoto, Japan) gas chromatograph, fitted with a flame ionization detector, was equipped with a 60-m SP-2560 fused silica capillary famewax column (0.25 mm i.d. × 0.25 μm film thickness) (Supelco, Inc., Bellefonte, PA, USA). The oven temperature was set at 190 °C for 2 min, then increased 5 °C/min to 250 °C and maintained for 5 min. The injector and detector temperatures were 255 °C. Standard fatty acids (Nu-Chek-Prep., Elysian, MN, USA) were run first to create a calibration reference.

### 4.7. Identification of Fatty Acid Desaturases from Soybean and Other Plant Species

The nucleotide and amino acid sequences of putative soybean fatty acid desaturases were retrieved from the soybean reference genome (*Glycine max*, Wm82.a2.v1) at Phytozome (v12.1) (https://phytozome.jgi.doe.gov). The fatty acid desaturases from other plant species, including *Arabidopsis thaliana*, *Phaseolus vulgaris* (v2.1), *Medicago truncatula* (Mt4.0v1), *Linum usitatissimum* (v1.0), *Oryza sativa* (v7_JGI), *Olea europaea* var. *sylvestris* (v1.0), and *Lotus japonicas* genome assembly build 3.0 (http://www.kazusa.or.jp/lotus/) were identified by BLASTP searches using the soybean fatty acid desaturase protein sequences as queries against the corresponding references of [79]. As a result, a total of 90 identified protein sequences with accession numbers were used in this study.

### 4.8. Phylogenetic Analysis

Multiple sequence alignments of the full-length protein sequences from nine plant species were performed by MUltiple Sequence Comparison by Log-Expectation (MUSCLE). An unrooted phylogenetic gene tree was then constructed by maximum likelihood (ML) method in MEGA X using the Jones–Taylor–Thornton Gamma Distributed (JTT+G) model with a bootstrap analysis of 1000 replicates [80,81].

### 4.9. Gene Structure and Expression Analysis

The genomic and coding sequences of soybean fatty acid desaturases retrieved from Phytozome v12.1 were aligned to generate the gene exon-intron structure diagram using the Gene Structure Display Server [82]. The normalized expression profile of soybean fatty acid desaturases was downloaded from RNA-Seq Atlas at SoyBase from six different tissues, including leaf, flower, pod, developmental seed, root and nodule (https://www.soybase.org/soyseq/). The results were used to generate heatmap and hierarchical clustering using an expression-based heat map [83].

### 4.10. Chromosomal Distribution and Syntenic Analysis

The locations of the soybean fatty acid desaturase genes and their corresponding chromosomes were drawn based on the soybean genome annotation a2.v1 on SoyBase. Syntenic analysis were performed using soybean fatty acid desaturases as locus identifiers in the plant genome duplication database (PGDD) [84] (http://chibba.agtec.uga.edu/duplication/) Nonsynonymous (Ka) versus synonymous substitution (Ks) rates were calculated based on their values retrieved from PGDD. Given the Ks values and a rate of 6.1 × 10^−9^ substitutions per site per year, the divergence time (T) was equal to Ks/(2 × 6.1 × 10^−9^) × 10^−6^ million years ago (Mya) for each gene pair [85].

### 4.11. In Silico Analysis

Multiple sequence alignments were performed using MUSCLE, in which the fatty acid desaturase mutations were overlaid on the amino acid sequence of Forrest wild type. Catalytic residues in conserved motifs of soybean fatty acid desaturases were identified from NCBI Conserved Domain Database (https://www.ncbi.nlm.nih.gov/cdd).

### 4.12. Homology Modeling of GmSACPD-C

The protein sequence of Forrest cv. GmSACPD-C was subjected to modeling with Deepview and Swiss Model Workspace software, as shown previously [11]. Mutation mapping and visualizations were performed using the UCSF Chimera package [86].

## 5. Conclusions

As the mainstream of gene functional analysis, complementation by gene transformation is still time-consuming and a costly process despite the efforts by the scientific community to establish the services-oriented transformation center. Recently, gene editing using CRISPR has emerged as a promising technology for functional gene analysis in soybeans; however, it is still very limited in its use and requires highly trained laboratories and focus only on few genes. Therefore, there is a strong need for new technologies combining high throughput screening methods while keeping lower cost for gene functional characterization. In this study, TbyS^+^, a versatile extension of the conventional TbyS and a high throughput technology coupled with bioinformatic tools to identify population-wide mutations, proved to be the method of choice to study gene function in soybean. TbyS^+^ allows for a fast identification of causal mutations based on target capture sequencing enrichment of selected genes, including promoters, exons, introns or any other regions of interest within a variety of sequenced genomes. This technology can be used efficiently to target and distinguish between gene family members that share in general high similarities and also genes with high copy numbers. We were able to discriminate mutations between members of each fatty acid desaturase gene family using capture sequencing enrichment and target recovery technology prior to next-generation sequencing. Meanwhile, large number of novel mutants in gene of interest were uncovered and became immediately available for soybean community.

## Figures and Tables

**Figure 1 ijms-22-04219-f001:**
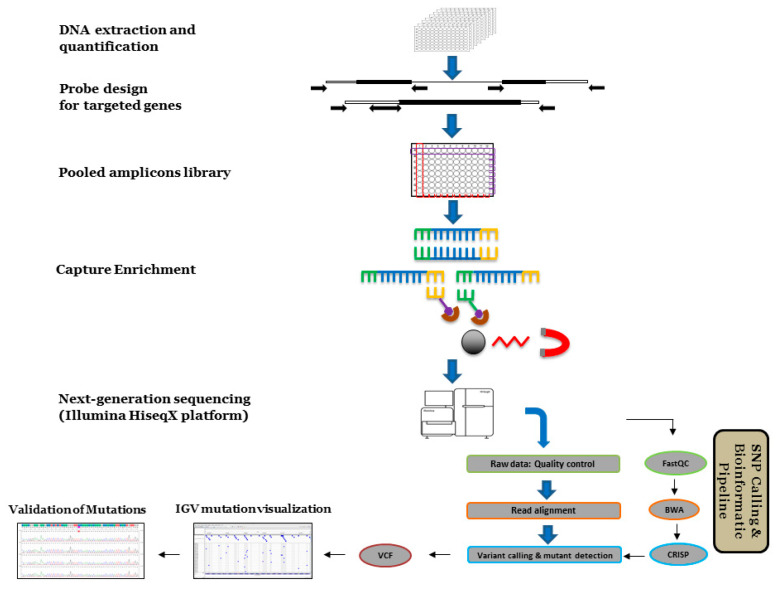
A scheme of TILLING by Sequencing^+^ (TbyS^+^). 4032 genomic DNA were extracted and stored in 96-well plates in −80 °C. A total of 42 plates of DNA samples were then pooled into three plates as a mutant pool library. For each target gene, a series of probes were designed to cover the whole gene. The amplicon sequencing was performed using Illumina HiSeq X platform. A custom bioinformatic pipeline was carried out to process the raw sequencing data and mutation identification. Individual mutant lines were finally confirmed by Sanger sequencing.

**Figure 2 ijms-22-04219-f002:**
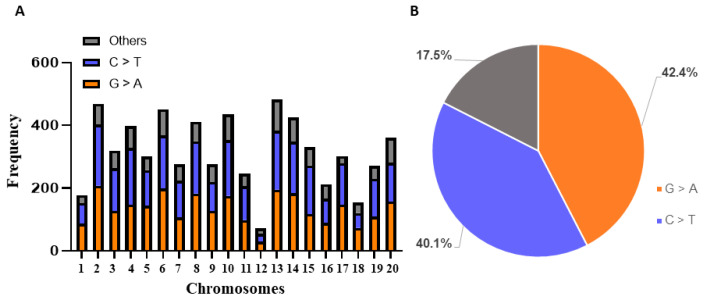
Characterization of induced mutations identified by TbyS^+^ in 138 soybean genes. (**A**) The distribution of mutation types in 20 soybean chromosomes. C > T and G > A represent typical EMS-type mutations; Others indicate non-EMS-type mutations. (**B**) The percentage of mutation types in 138 soybean genes.

**Figure 3 ijms-22-04219-f003:**
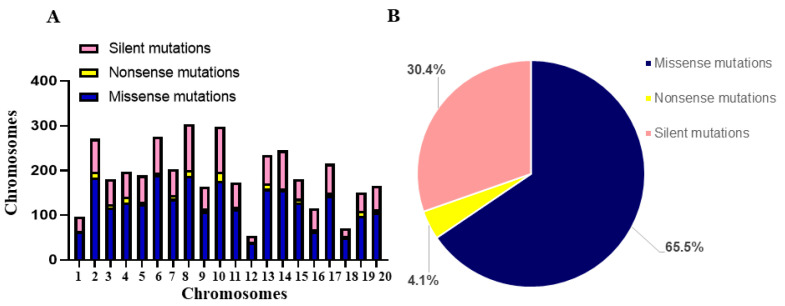
Characterization of mutation effects in 138 soybean genes. (**A**) The distribution of mutation effects in 20 soybean chromosomes. (**B**) The percentage of mutation effects in 138 soybean genes.

**Figure 4 ijms-22-04219-f004:**
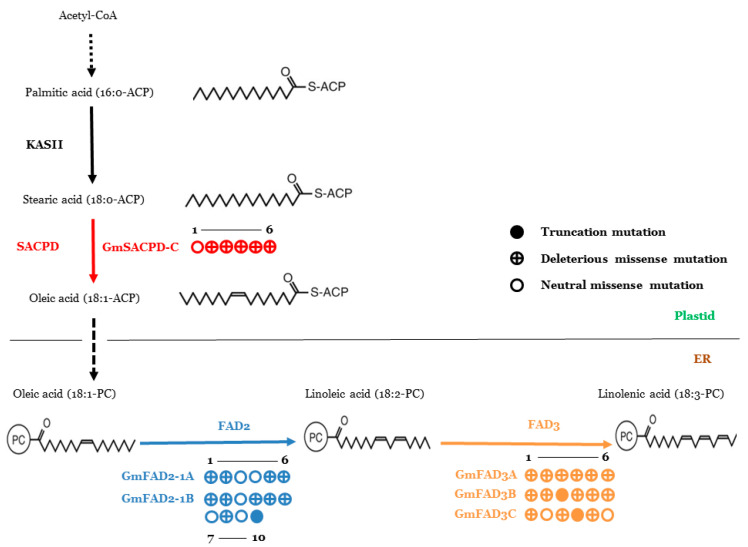
Summary of induced mutations that were identified by TbyS^+^ in six soybean desaturases. Six mutations are shown for each gene. The deleterious and neutral missense mutations were predicted by PROVEAN.

**Figure 5 ijms-22-04219-f005:**
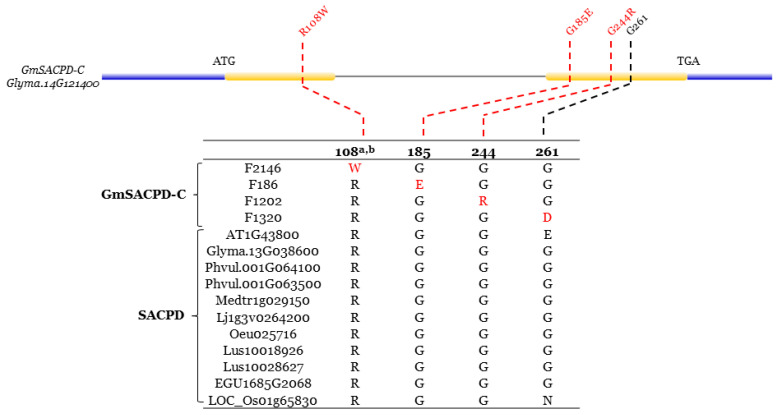
Conserved residue variations in the stearoyl-acyl carrier protein desaturase (SACPD). Protein sequence alignment of four GmSACPD-C mutants and SACPD orthologs from other eight plant species are partially shown to illustrate the amino acid difference resulting from the nucleotide polymorphisms in the predicted protein sequences. For the identified *GmSACPD-C* mutants, amino acid differences at conserved residues were linked with nucleotide polymorphisms using the red dotted lines, whereas the black dotted lines represented amino acid changes at non-conserved residues. a. Numbering of the residues are according to the sequence of GmSACPD-C. b. residue at di-iron center. Red amino acids show the EMS induced mutations found in the TILLING mutants.

**Figure 6 ijms-22-04219-f006:**
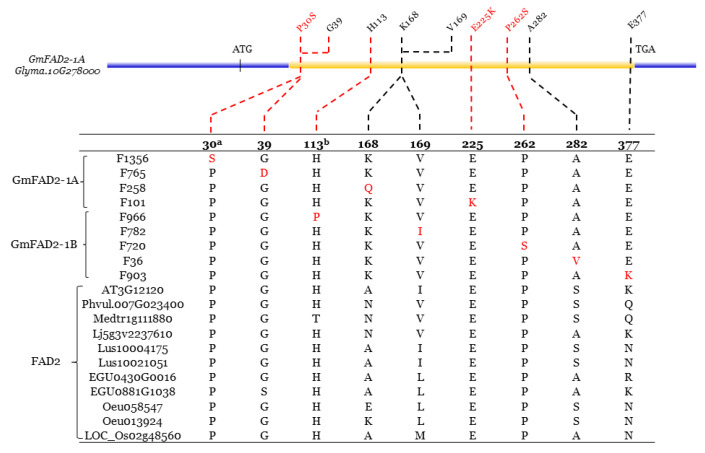
Conserved residue variations in the fatty acid desaturases-2 (FAD2). The amino acid difference in nine GmFAD2-1A/1B mutants and FAD2 orthologs from other eight plant species are partially shown in the table where the number on top indicates the position of an amino acid in the predicted protein. For the identified GmFAD2-1A/1B mutants, amino acid differences at conserved residues were linked with nucleotide polymorphisms using the red dotted lines, whereas the black dotted lines represented amino acid changes at non-conserved residues. a. Numbering of the residues is according to the sequence of GmFAD2-1A. b. Residue at a putative di-iron ligand.

**Figure 7 ijms-22-04219-f007:**
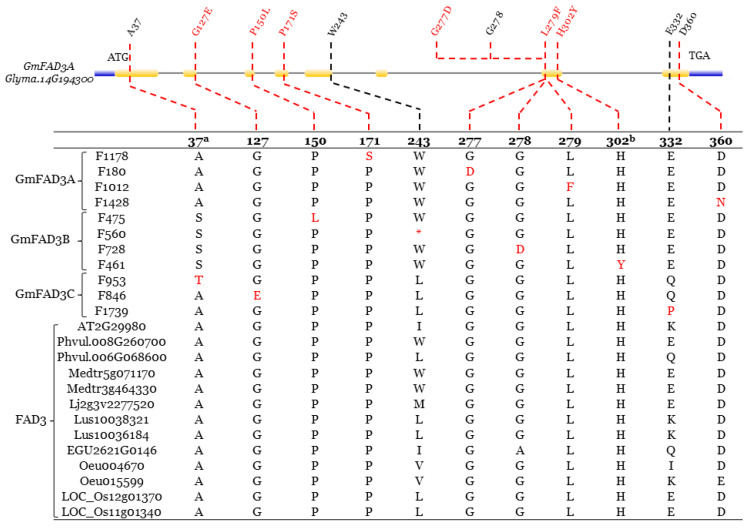
Conserved residue variations in fatty acid desaturases-3 (FAD3). Conserved residue variations among omega-3 desaturases. Protein sequence alignment of 11 GmFAD3A/C mutants and omega-3 desaturase orthologs from other eight plant species are partially shown to illustrate the amino acid difference resulting from the nucleotide polymorphisms in the predicted protein sequences. For the identified GmFAD3A/C mutants, amino acid differences at conserved residues were linked with nucleotide polymorphisms using the red dotted lines, whereas the black dotted lines represented amino acid changes at non-conserved residues. a. Numbering of the residues is according to the sequence of GmFAD3-A. b. Residue at a putative di-iron ligand.

**Figure 8 ijms-22-04219-f008:**
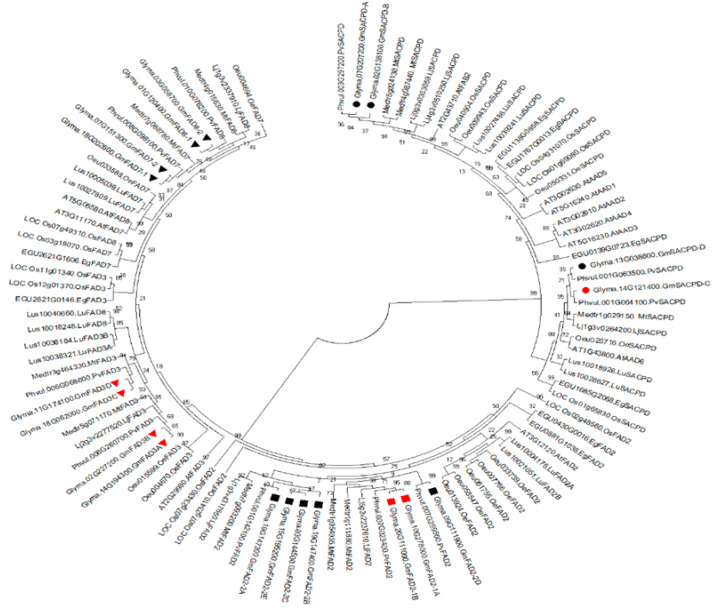
Phylogenetic gene tree of fatty acid desaturases from nine plant species. The protein sequences of all fatty acid desaturases were subjected to a MUSCLE alignment and phylogenetic gene tree was constructed using Mega X. The name and abbreviation of plant species used for the analysis are: *Arabidopsis thaliana* (At); *Glycine max* (Gm); *Phaseolus vulgaris* (Pv); *Medicago truncatula* (Mt); *Lotus japonicas* (Lj); *Elaeis guineensis* (Eg); *Oriza sativa* (Os); *Linum usitatissimum* (Lu); *Olea europaea* (Oe). Members of GmSACPD, GmFAD2, and GmFAD3 are marked with circle, square, and triangle, respectively, from which seven fatty acid desaturases controlling soybean fatty acid content are highlighted in red.

**Figure 9 ijms-22-04219-f009:**
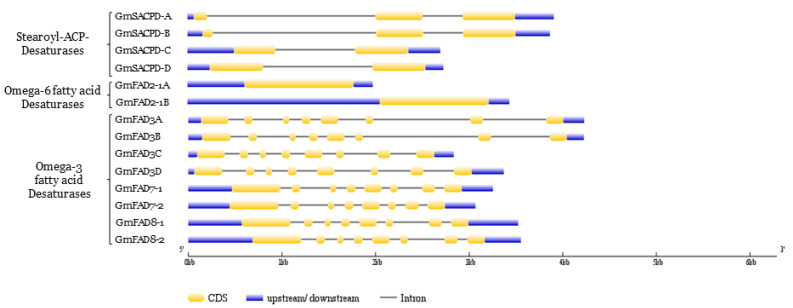
Gene structure of soybean fatty acid desaturases from three gene families. The structures of 14 soybean fatty acid desaturases genes were plotted with yellow boxes representing exons (coding DNA sequence, CDS), black lines illustrating introns, and blue boxes indicating 5′-UTR and 3′-UTR regions. The size of gene structures could be measured by the scale in the unit of base pair (bp) at the bottom. The gene structures were drawn using the Gene Structure Display Server.

**Figure 10 ijms-22-04219-f010:**
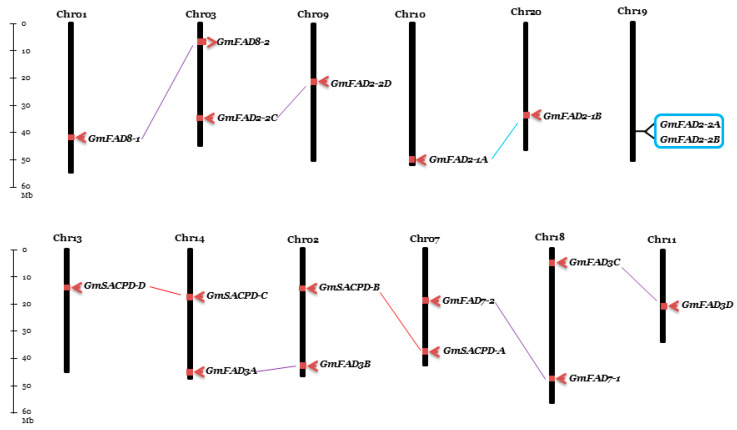
Chromosomal locations and duplications of the 19 soybean fatty acid desaturase genes. Each chromosome number is indicated above the bar by Roman number and the scale (on the left) is in mega base (Mb). The size of chromosome and gene locations are based on soybean genome annotation a2.v1 on SoyBase. Each pair of segmental duplication in *GmSACPD*, *GmFAD2*, and *GmFAD3* is connected by red, blue, and purple lines, respectively. The tandem duplicated genes are shown in the rectangular box.

**Table 1 ijms-22-04219-t001:** A summary of mutations in six fatty acid desaturase genes identified by TbyS^+^.

Gene ID	Amplicon Size (bp)	Base Changes	Type of Base Changes			Amino Acid Substitutions	Missense Mutations	Nonsense Mutations	Silent Mutations
			G > A	C > T	Others				
*GmSACPD-C*	1800	45	21	20	4	38	27	0	11
*GmFAD2-1A*	1920	42	13	24	5	26	16	0	10
*GmFAD2-1B*	3320	62	17	22	23	24	10	1	13
*GmFAD3A*	2480	45	20	15	10	17	12	0	5
*GmFAD3B*	2480	44	23	16	5	22	14	3	5
*GmFAD3C*	2440	36	17	10	9	20	13	1	6

**Table 2 ijms-22-04219-t002:** Fatty acid phenotypes of the selected mutants in soybean fatty acid desaturase genes identified by TbyS^+^.

Gene ID	Plant ID	Nucleotide Change	Amino Acid Substitution	16:0	18:0	18:1	18:2	18:3
*GmSACPD-C*	F2146	C322T	R108W	9.6	**11.7**	18.1	50.7	9.8
	F186	G554A	G185E	9.4	**5.6**	23.1	53.7	6.3
	F1202	G730A	G244R	10.5	**6.0**	20.5	52.0	8.8
	F1320	G782A	G261D	10.4	**5.3**	23.1	54.9	4.4
*GmFAD2-1A*	F1356	C88T	P30S	11.0	3.7	**25.0**	51.2	7.4
	F765	G116A	G39D	10.0	3.8	**27.0**	50.2	7.5
	F258	A502C	K168Q	12.2	5.6	**34.5**	39.3	4.5
	F101	G673A	E225K	10.6	4.6	**29.6**	45.9	7.7
*GmFAD2-1B*	F966	A338C	H113P	10.7	4.1	**23.0**	54.1	6.6
	F782	G505A	V169I	11.7	4.9	**27.8**	40.2	5.3
	F720	C784T	P262S	10.7	5.1	**26.2**	50.2	6.2
	F36	C845T	A282V	10.9	4.9	**27.6**	46.4	8.5
	F903	G1129A	E377K	12.4	5.0	**25.1**	45.9	7.7
*GmFAD3A*	F1178	C511T	P171S	11.2	4.5	19.0	57.7	**5.8**
	F180	G830A	G277D	17.2	5.5	20.1	45.0	**5.2**
	F1012	C835T	L279F	10.4	3.5	34.2	45.7	**4.5**
	F1428	G1078A	D360N	10.1	5.1	21.1	56.1	**5.6**
*GmFAD3B*	F475	C461T	P154L	10.0	4.3	26.5	53.3	**4.6**
	F560	G741A	W247 *	10.4	3.3	18.9	60.5	**5.5**
	F728	G845A	G282D	12.2	4.2	25.2	52.3	**4.6**
	F461	C916T	H306Y	12.5	4.6	17.4	53.1	**4.9**
*GmFAD3C*	F953	G112A	A38T	11.2	3.2	23.2	54.9	**5.7**
	F846	G383A	G128E	13.9	5.0	17.0	39.6	**4.8**
	F1739	A998C	Q333P	9.4	4.5	37.1	41.8	**5.1**
	**F-WT**			**10.8**	**3.8**	**20.0**	**56.2**	**7.2**

* The highlighted fatty acid levels represent the highest or lowest content obtained from each M2 line. Bold shows the improved/increased oil content in question.

## Data Availability

Not applicable.

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
