# Peer review of "TILLING-by-Sequencing+ to Decipher Oil Biosynthesis Pathway in Soybeans: A New and Effective Platform for High-Throughput Gene Functional Analysis"

_ijms, 2021, doi:10.3390/ijms22084219_

Round 1

Reviewer 1 Report

The authors analyzed soybean mutants by NGS-based approach and detected mutations in oil biosynthetic genes. The ms is basically well-written and well-presented. Therefore, I would recommend this ms for publication in IJMS. However, it needs numerous corrections prior to the publication as pointed out in the attached pdf file.

Author Response

Reviewer 1

Comments and Suggestions for Authors

The authors analyzed soybean mutants by NGS-based approach and detected mutations in oil biosynthetic genes. The ms is basically well-written and well-presented. Therefore, I would recommend this ms for publication in IJMS. However, it needs numerous corrections prior to the publication as pointed out in the attached pdf file.

Answer: We have taken into consideration all suggestions and comments of reviewer 1. The edits can be found in track change in the attached edited version of the manuscript.

The authors would like to thank reviewer 1 for all his comments and suggestions to improve the manuscript.

Reviewer 2 Report

The MS entitled "TILLING-by-sequencing+ to decipher oil biosynthesis pathway in soybeans: A new and effective platform for high-throughput gene functional analysis" is very interesting to the soybean community. Authors have performed Tilling by sequencing and identified some novel mutations in oil biosynthesis pathway genes. The MS is well designed and written, however, there are some minor edits that need to be addressed.

Line-154: The clean fastq reads were mapped to Williams 82 reference genome- Please add the version of the genome and from which site genome was downloaded

Line160-Which reference genome was selected to identify the functional prediction of SNPs-was the genome same used for alignment and downloaded from ensemble-if the reference genome is downloaded from phytozome and snpeffect from ensemble then authors need to check the exact position of the SNPs, are matching or not.

Section 3.2 is a completely MM section- need to add more results than methods

Line 399-Typo- Table 1014. bp (GmSACPD-C) to 1209 bp (GmSACPD-A) with an average of 1134 bp.-Please check carefully

typo-Scheme 48. N, R108W, G185E, A238V, G244R and G261D) were detected

Figure 1 and Figure 10 need improvement

Formatting of tables and figures is required

The authors have not discussed heterozygosity, how it was addressed. What about the large deletions?

Author Response

Reviewer 2

Comments and Suggestions for Authors

The MS entitled "TILLING-by-sequencing+ to decipher oil biosynthesis pathway in soybeans: A new and effective platform for high-throughput gene functional analysis" is very interesting to the soybean community. Authors have performed Tilling by sequencing and identified some novel mutations in oil biosynthesis pathway genes. The MS is well designed and written, however, there are some minor edits that need to be addressed.

Line-154: The clean fastq reads were mapped to Williams 82 reference genome- Please add the version of the genome and from which site genome was downloaded

Answer: We have added the WI82 reference genome version (WI82.a2.v1) that was used to map the fastq reads.

Line160-Which reference genome was selected to identify the functional prediction of SNPs-was the genome same used for alignment and downloaded from ensemble-if the reference genome is downloaded from phytozome and snpeffect from ensemble then authors need to check the exact position of the SNPs, are matching or not.

Answer: The same reference genome version, WI82.a2.v1, was used for SNP calling and mutation effect prediction. We have checked the exact position of each SNP. Furthermore, most of the TILLING mutants that were isolated have been target sequenced (Sanger sequencing) to confirm the exact mutations and positions as shown in Table 2, Figures 5, 6, 7 and Tables S3 and S4.

Section 3.2 is a completely MM section- need to add more results than methods

Answer: In addition to the functional characterization of genes at the fatty acid biosynthesis pathway, this manuscript describes the development of TILLING by Sequencing+, a versatile extension of the conventional TbyS and a high throughput technology coupled with bioinformatic tools to identify population-wide mutations. Therefore, describing the developed TILLING by sequencing+ methodology in this section (3.2) will be very useful to the broad scientific audience interested in adopting this technology. 

Line 399-Typo- Table 1014bp (GmSACPD-C) to 1209 bp (GmSACPD-A) with an average of 1134 bp.-Please check carefully

Answer: Typo in line 399 has been corrected.

typo-Scheme 48. N, R108W, G185E, A238V, G244R and G261D) were detected

Answer: Typo has been corrected.

Figure 1 and Figure 10 need improvement

Answer: Figures 1 and 10 were improved as suggested by the reviewer.

Formatting of tables and figures is required

Answer: Most of the figures and Tables were organized as suggested by the reviewer.

The authors have not discussed heterozygosity, how it was addressed. What about the large deletions?

EMS is a random mutagen that allow the generation of random SNPs in the genome. Most of the mutations that were present in the oil biosynthesis genes due to the EMS mutagenesis are SNPs, with G > A and C > T being the frequently identified type of base changes that were present at high percentages: 91%, 88%, 62%, 77%, 88%, 75% for GmSACPD-C, GmFAD2-1A, GmFAD2-1B, GmFAD3A, GmFAD3B, and GmFAD3C, respectively. Unlike other mutagens like sodium azide or fast neutron, EMS mutagenesis does not generate large deletions. At the M2 generation, most of the mutations were heterozygous. Therefore, the seeds were advanced to the M3 generation, we observed the following mutation segregations, 25% revert to the wild type, 50% are heterozygous, and 25% are homozygous at the identified mutations. The M3 mutants were sanger sequenced to validate the homozygous mutations for further phenotyping (Lines 595-604).

The authors would like to thank the reviewer 2 for all his constructive comments and suggestions to improve the manuscript.